



# Technical note: Determination of binary gas phase diffusion coefficients of unstable and adsorbing atmospheric trace gases at low temperature – Arrested Flow and Twin Tube method

Stefan Langenberg[1,a], Torsten Carstens[1,2], Dirk Hupperich[1], Silke Schweighoefer[1,b], and Ulrich Schurath[3]

[1]Institut für Physikalische und Theoretische Chemie, University of Bonn, Bonn, Germany
[2]Karlsruher Institut für Technologie, Karlsruhe, Germany
[3]Institut für Umweltphysik, University of Heidelberg, Heidelberg, Germany
[a]now at: Klinik und Poliklinik für Hals-Nasen-Ohrenheilkunde/Chirurgie, University of Bonn, Bonn, Germany
[b]now at: UP GmbH, Ibbenbüren, Germany

**Correspondence:** Stefan Langenberg (langenberg@uni-bonn.de)

**Abstract.** Gas phase diffusion is the first step for all heterogeneous reactions under atmospheric conditions. Knowledge of binary diffusion coefficients is important for the interpretation of laboratory studies regarding heterogeneous trace gas uptake and reactions. Only for stable, nonreactive and non polar gases well-established models for the estimation of diffusion coefficients from viscosity data do exist. Therefore, we have used two complementary methods for the measurement of binary diffusion
coefficients in the temperature range of 200 K to 300 K: the arrested flow method is best suited for unstable gases and the twin tube method is best suited for stable but adsorbing trace gases. Both methods were validated by measurement of diffusion coefficients of methane and ethane in helium and air and nitric oxide in helium. Using the arrested flow method the diffusion coefficients of ozone in air, dinitrogen pentoxide and chlorine nitrate in helium and nitrogen were measured. The twin tube method was used for the measurement of the diffusion coefficient of nitrogen dioxide and dinitrogen tetroxide in helium and
nitrogen.

## 1 Introduction

The critical role of heterogeneous reactions in atmospheric chemistry is widely accepted. The diffusion of gas molecules towards the surface is the first step in a heterogeneous reaction, and it can influence and sometimes even control the overall rate
of the uptake of a trace gas onto the surface (Kolb et al., 2010; Tang et al., 2014a). Diffusion also plays a role in atmosphere – biosphere interactions: the incorporation of trace gases like ozone and nitrogen dioxide into leaves and isoprene out through stomata is diffusion controlled (Laisk et al., 1989; Eller and Sparks, 2006; Fall and Monson, 1992).





**Table 1.** Lennard-Jones parameters of the species investigated in this study

| Species | Formula | M [g mol$^{-1}$] | $\sigma$ [Å] | $\epsilon/k$ [K] | Method | Source |
|---|---|---|---|---|---|---|
| Helium | He | 4.00 | 2.551 | 10.2 | v | Poling et al. (2004) |
| Nitrogen | N$_2$ | 28.01 | 3.798 | 71.4 | v | Poling et al. (2004) |
| Air | | 28.81 | 3.711 | 78.6 | v | Poling et al. (2004) |
| Methane | CH$_4$ | 16.04 | 3.758 | 148.6 | v | Poling et al. (2004) |
| Ethene | C$_2$H$_4$ | 28.05 | 4.163 | 224.7 | v | Poling et al. (2004) |
| Nitric oxide | NO | 30.01 | 3.492 | 116.7 | v | Poling et al. (2004) |
| Nitrogen dioxide | NO$_2$ | 46.01 | 3.765 | 210.0 | v | Brokaw and Svehla (1966) |
| Ozone | O$_3$ | 48.00 | 3.875 | 208.4 | b | Massman (1998) |
| Dinitrogen tetroxide | N$_2$O$_4$ | 92.01 | 4.621 | 347.0 | v | Brokaw and Svehla (1966) |
| Chlorine nitrate | ClONO$_2$ | 97.46 | 4.470 | 364.7 | b | Patrick and Golden (1983) |
| Dinitrogen pentoxide | N$_2$O$_5$ | 108.01 | 4.570 | 450.0 | b | Patrick and Golden (1983) |

v = obtained from viscosity data, b = obtained from $T_b$ and $V_b$ using Eqns. (4).

Marrero and Mason (1972); Massman (1998); Tang et al. (2014a, 2015) and Gu et al. (2018) compiled and evaluated the available experimental data of diffusion coefficients of atmospheric trace gases. However, the existing compilations focus on stable gases, experimental diffusion coefficients of ozone, nitrogen dioxide, chlorine nitrate and dinitrogen pentoxide are still missing. They cannot be predicted with the required accuracy because detailed kinetic theory requires intermolecular potentials which are not generally available for atmospherically relevant compounds.

Chapman and Enskok derived the following equation from kinetic theory of gases for the molecular binary diffusion coefficient

$$D = \frac{3}{16}\sqrt{\frac{2\pi kT(m_A + m_B)}{m_A m_B}}\left(\frac{kT}{\pi\sigma_{AB}^2\Omega_D p}\right) \tag{1}$$

where $m$ is the mass of the molecules, $k$ is the Boltzmann constant, $p$ the pressure and $T$ is the absolute temperature. $\sigma_{AB}$ is the characteristic length of the intermolecular force law, $\Omega_D$ is the dimensionless collision integral of diffusion. It depends on the temperature and the characteristic energy $\epsilon_{AB}$ of the Lennard-Jones potential describing the intermolecular force (Poling et al., 2004; Marrero and Mason, 1972). $\Omega_D$ as function of temperature is expressed by the fit function

$$\Omega_D = \frac{A}{\Theta^B} + \frac{C}{\exp(D\Theta)} + \frac{E}{\exp(F\Theta)} + \frac{G}{\exp(H\Theta)} \tag{2}$$

where $\Theta = kT/\epsilon_{AB}$, $A = 1.06036$, $B = 0.15610$, $C = 0.19300$, $D = 0.47635$, $E = 1.03587$, $F = 1.52996$, $G = 1.76474$, $H = 3.89411$ (Neufeld et al., 1972; Poling et al., 2004). The equations

$$\epsilon_{AB} = \sqrt{\epsilon_A\epsilon_B}, \qquad \sigma_{AB} = \frac{\sigma_A + \sigma_B}{2} \tag{3}$$





are usually employed to relate the interaction parameters of the Lennard-Jones potential between components A and B to the

interaction potential parameters of the individual components. A tabulation of the potential parameters of the species considered

in this work is given in Table 1. The Lennard-Jones parameters $\sigma$ and $\epsilon$ are generally not available for unstable atmospheric

trace gases. Patrick and Golden (1983) estimated them by the equations

$$\sigma = 1.18\, V_b^{1/3}, \qquad \epsilon/k = 1.21\, T_b \tag{4}$$

from $T_b$ the normal boiling point temperature and $V_b$ the molar volume at boiling point. In cases where $V_b$ cannot be determined

experimentally, it is obtained from tables of atomic volumes using the LeBas method. Patrick and Golden (1983) assumed the

systematic errors of $\sigma$ and $\epsilon$ obtained by this method to be $\leq 20\%$.

The diffusion coefficient as function of pressure in a narrow temperature range close to the reference temperature $T_0$ is

usually expressed as

$$D = D_0 \left( \frac{p_0}{p} \right) \left( \frac{T}{T_0} \right)^b \tag{5}$$

where $T_0 = 273.15$ K are the standard temperature and $p_0 = 101325$ Pa the standard pressure (STP). Close to the reference

temperature $T_0$, the temperature coefficient $b$ can be calculated as follows (Poling et al., 2004)

$$b = \left( \frac{\partial \ln D}{\partial \ln T} \right) = \frac{3}{2} - \left( \frac{\partial \ln \Omega_D}{\partial \ln T} \right) = \frac{3}{2} - \frac{T_0}{\Omega_D} \left( \frac{\partial \Omega_D}{\partial T} \right), \tag{6}$$

$$b = \frac{3}{2} - \frac{\Theta}{\Omega_D} \left( \frac{\partial \Omega_D}{\partial \Theta} \right) \tag{7}$$

From Eq. (2) it is obtained by derivation

$$\left( \frac{\partial \Omega_D}{\partial \Theta} \right) = -\frac{AB}{\Theta^{B+1}} - \frac{CD}{\exp(D\Theta)} - \frac{EF}{\exp(F\Theta)} - \frac{GH}{\exp(H\Theta)} \tag{8}$$

Fuller et al. (1966) developed a simple correlation equation for the estimation of gas phase diffusion coefficients using

additive atomic volumes $V_A$ and $V_B$ for each species. With the molar masses $M_A$ and $M_B$ ($[M] = \mathrm{g\,mol^{-1}}$) of each species

and $[p] = $ bar, the diffusion coefficient ($[D] = \mathrm{cm^2\,s^{-1}}$) is given by

$$M_{AB} = \frac{2}{1/M_A + 1/M_B} \tag{9}$$

$$D = 0.00143 \frac{T^{1.75}}{\sqrt{M_{AB}}(V_A^{1/3} + V_B^{1/3})^2 p} \tag{10}$$

Tabulations of atomic volume increments are summarized by Poling et al. (2004) and Tang et al. (2014a).

In the atmosphere, for typical submicron-sized aerosol particles, gas-phase diffusion does not usually limit uptake. Therefore,

for modelling atmospheric processes, it is sufficient to use diffusion coefficients obtained using the Fuller method. However,

in many laboratory experiments for the measurement of mass accommodation coefficients, conditions are such that gas-phase

diffusion limitations need to be taken into account (Kirchner et al., 1990; Müller and Heal, 2002; Davidovits et al., 2006).



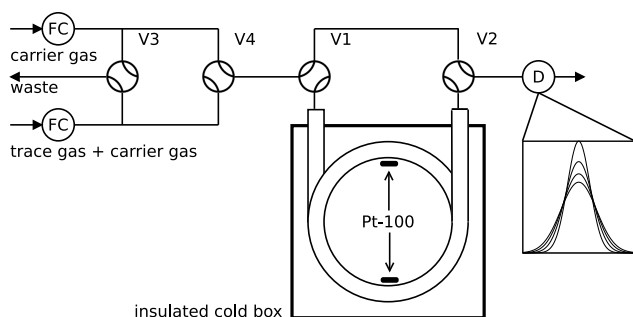

**Figure 1.** Arrested flow method: a pulse of trace gas is introduced into the column by simultaneously switching valves V3 and V4 for a short time. After the peak has reached the middle of the column, the carrier gas is bypassed by switching V1 and V2 for arrest times $t_a$ between 0 and 200 s. After each arrest time $t_a$ the corresponding peak shape is recorded by the detector D. The gas flow is controlled by flow controller FC.

## 2 Methods

### 2.1 Arrested flow method

The arrested flow (AF) method was first described by Knox and McLaren (1964) and McCoy and Moffat (1986): the diffusion coefficient of a given trace gas is derived from the broadening of width $\varsigma_t$ of trace gas plugs arrested for different times in a long void gas chromatography glass column (length $l = 2.8$ m, radius $r = 0.189$ cm). A plug is generated by injecting a small amount of dilute trace gas into a steady flow of carrier gas by means of computer controlled solenoid valves. The flow is arrested when the plug has travelled halfway down the tube, see Fig. 1. In the absence of turbulence, the initial plug profile

spreads out along the tube by molecular diffusion only. Until the flow is arrested, the box profile of the trace gas has reshaped to Gaussian by Taylor diffusion (Taylor, 1953, 1954), if the condition

$$l \gg \frac{\dot{V}}{\pi D} \tag{11}$$

is fulfilled, where $\dot{V}$ is the carrier gas flow rate. After a given arrest time $t_a$, the trace gas is eluted with $\approx 20$ sccm (1 sccm = 1 ml min$^{-1}$ at 273.15 K and 1013 hPa) and the concentration profile is measured with a suitable gas chromatography detector.

This procedure is repeated for different arrest times $t_a$. The experimental peak profiles are fitted to Gaussians to determine the peak variance $\varsigma_t^2$. According to theory based on Fick's second law of diffusion

$$\left(\frac{\partial c}{\partial t}\right)_z = D \left(\frac{\partial^2 c}{\partial z^2}\right)_t \tag{12}$$

a plot of $\varsigma_t^2$ versus arrest time $t_a$ should be linear. The slope of the plot of $\varsigma_z$ vs. $t_a$ is given by

$$\frac{\Delta \varsigma_z^2}{\Delta t_a} = 2D \tag{13}$$





Since the variance is measured in units of time, it has to be converted to units of length using the gas flow speed $v$ in the column

$$\Delta\varsigma_z^2 = v^2 \Delta\varsigma_t^2. \tag{14}$$

From the carrier gas mass flow $\dot{n}$, temperature $T$ and pressure $p$ in the column which approximately equals atmospheric pressure, the flow speed can be determined by

$$v = \frac{\dot{n}RT}{\pi r^2 p} \tag{15}$$

The column is embedded in an aluminum block which is cooled by a recirculating cryostat (Lauda RLS6). The aluminum block is mounted in a plastic box insulated by Styrodur. The column temperature homogeneity is monitored with 2 Pt-100 sensors connected to the upper and lower parts of the column coil. The solenoid valves are connected by 1/16" Teflon tubes and controlled by computer using the software Asyst 3.1 (Keithley). At each temperature, 12 to 20 peaks are recorded at different

arrest times.

The systematic error of the determined diffusion coefficients using this method primarily depends on the systematic error of measuring the inner diameter of the column and the systematic error of the mass flow rate. A Teflon tube pushed through the column was used to determine the length of the column. The void volume of the column was determined by filling the column with water and measuring the weight of the water. From volume and length, the cross-sectional area and radius are calculated,

yielding a mean radius with a systematic error of 0.5%. After the experiments, the column was cut into small fragments. The inner diameter of these fragments was measured using a caliper gauge. We found that the inner diameter synchronously changes with column winding with a variability of 1%. When using Eq. (14) to transform $\Delta\varsigma_t$ to $\Delta\varsigma_z$, not the mean velocity but the actual velocity $v$ and radius $r$ at the location where the peak is arrested are relevant. Therefore, the actual systematic error of the radius is about 1%. The mass flow controllers were calibrated using a soap bubble flow meter. Thus, the systematic

error of the mass flow rate is about 1.5%. This sums up to a total theoretical systematic error of the AF method to about 7%. The random error of the method is about $> 0.4\%$, twice the repeatability $> 0.2\%$ of the flow rate.

## 2.2 Twin tube method

The twin tube (TT) method is a steady state technique for diffusion coefficient measurements over a wide temperature range using a diffusion bridge (Marrero and Mason, 1972). It is insensitive to wall adsorption effects which may invalidate AF

measurements at low temperature. Our apparatus consists of two parallel horizontal flow tubes (length 2 m, inner diameter 10 mm) which are connected by a bunch of $n = 220$ carefully thermostatted fused silica capillaries of radius $r = (39.2 \pm 0.4)$ µm and length $l = (20.8 \pm 0.3)$ mm, see Fig. 2. The capillaries are embedded in a block made of brass. The cooling liquid of a cryostat (Lauda RLS6) circulates through the brass block, thereby covering the range from ambient temperature down to 198 K. Close to the diffusion bridge, the temperature in the block is measured with two Pt-100 sensors. The capillaries are pasted into

two parallel slits in a short section of the parallel flow tubes that is made of stainless steel. Up and downstream of the brass block, the flow tubes consist of glass. The entire apparatus consisting of the flow tubes and the diffusion bridge are housed





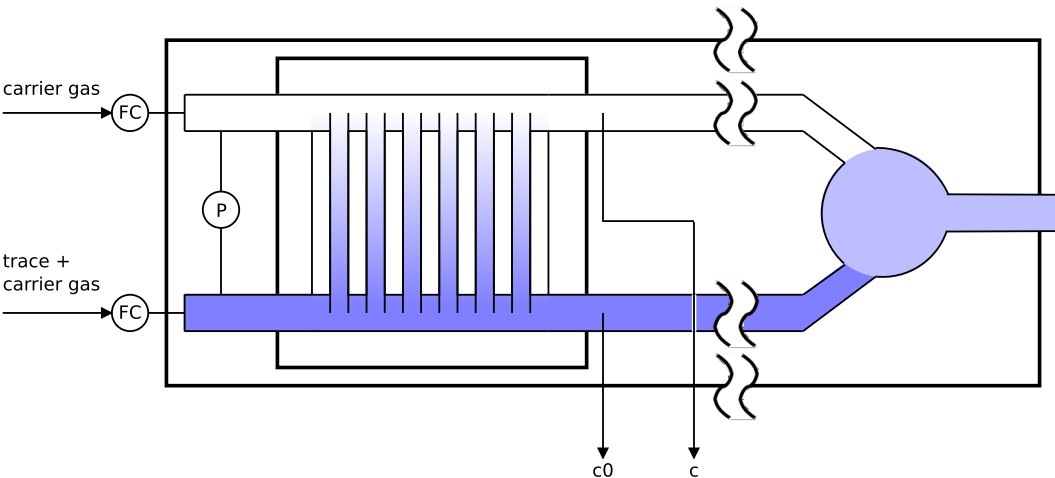

**Figure 2.** Schematic diagram of the twin tube experiment (false to scale): a diffusion bridge is connecting two flow tubes. Downstream the diffusion bridge some gas is continuously sampled for analysis to determine the trace gas concentrations $c_0$ and $c$.

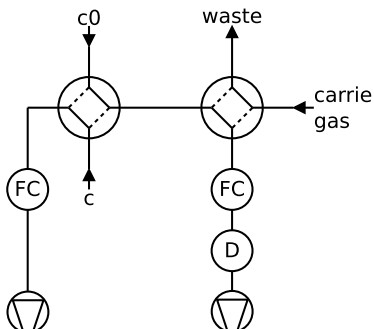

**Figure 3.** Twin tube method, continuous mode: a partial current is sucked through the detector D by a mass flow controller FC. A second 4-port valve enables to switch the detector to pure carrier gas to record the baseline.




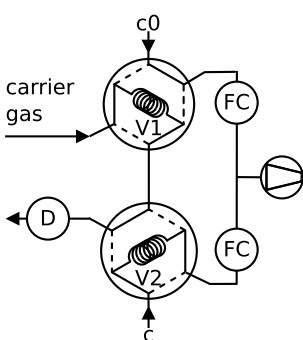

**Figure 4.** Twin tube method, peak mode: the carrier gas is admitted first through the sample loop V1 or V2 of a six-port valve. Then the content of the sample loop is pushed into the detector as peak after switching the six-port valve.

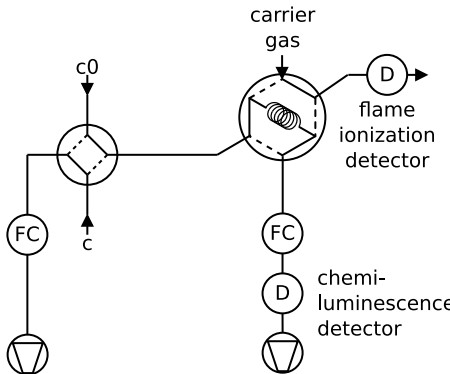

**Figure 5.** Twin tube method with internal standard as used for $NO_2$ in $N_2$: the species under investigation is monitored in continuous mode, the internal standard is sampled by a 6-port valve and detected by a flame ionization detector.

in a large insulated box which can be cooled down to 260 K. After changing the setting of the recirculating thermostat by an increment of 10 K it takes about 1 h until the temperature of the diffusion bridge has equilibrated.

Pure carrier gas is flown through one of the flow tubes, while a constant trace gas concentration $c_0$ is maintained in the

other. A concentration gradient is established along the capillaries. This gives rise to a constant flux $J_D$ by molecular diffusion through the diffusion bridge described by Fick's first law of diffusion

$$J_D = -D \left( \frac{\partial c}{\partial z} \right)_t = D \left( \frac{c_0 - c}{l} \right) \tag{16}$$

where $c$ is the trace gas concentration at the low concentration end. Pressure differences between the flow tubes are carefully eliminated to suppress trace gas transport by viscous flow through the capillaries. This requires that both flow tubes are totally

symmetric. The difference pressure is monitored using a differential high accuracy pressure transducer (MKS model 398, measuring range $10^{-4}$ Torr to 1 Torr). By measuring the concentration ratio in both flow tubes, downstream the diffusion





bridge the diffusion coefficient is obtained by

$$D = \frac{\dot{V}l}{n\pi r^2} \frac{c}{c_0} \tag{17}$$

when $c \ll c_0$ where $\dot{V}$ is the volume flow rate of the carrier gas. The ratio of mass transport by viscous flow to diffusion flow

through the capillaries is given by (viscosity $\eta$)

$$\frac{J_V}{J_D} \approx \frac{r^2 \Delta p}{16 \eta D}. \tag{18}$$

Therefore, the ratio of interfering viscous trace gas to mass flow by diffusion was minimized by using narrow capillaries. For the diffusion of $NO_2$ in $N_2$ at standard pressure and temperature, the fraction of viscous flow can be held $< 1\%$ when keeping the differential pressure $\Delta p < 2 \times 10^{-4}$ Torr. During the TT-experiments the differential pressure was maintained low so that

the fraction of viscous flow was less than $0.3\%$.

A trace gas detector is required which is linear over a wide concentration range and stable over time. Depending on the trace gas and the detector properties, the trace gas can either be detected by continuous mode (Fig. 3) or by peak mode (Fig. 4). The low concentration is determined with a random error of about $1\%$. The signal of the trace gas detector is fed into a A/D-converter with 16-bit resolution (Data Translation DT2705/5715A).

The supplier of the capillary columns used in this work as raw material for assembling the diffusion bridge reports the inner diameter with a systematic error of $10\%$. This is too much for the measurement of diffusion coefficients. Therefore, two segments of the column were used to determine the inner diameter by weighing an empty and a water filled section of the capillary column. Thereby the radius of the diffusion bridge capillaries was determined with a systematic error of $1\%$. We tried to validate the result by electron micrography of two cross sections of the column material. However, the systematic error of

the inner diameter measured by electron micrography is about $5\%$. When assuming a systematic error of the flow rate of $1.5\%$, this results in a total systematic error of the method of $< 4\%$. The random error of the method depends on the random error of about $1\%$ of the determination of the lower concentration $c$.

Later on during the experiments it was found that some capillaries of the diffusion bridge became blocked by dust or condensed matter. Fortunately, the TT method can be utilized to measure diffusion coefficients of several species simultaneously

when using the peak mode and a gas chromatograph as detector. If the diffusion coefficient of one of the trace gases has been determined with another reliable technique at one temperature, this diffusion coefficient can be used as internal standard, see Fig. 5. It is assumed that the effective area of the capillaries is independent of temperature.

## 3   Results and discussion

To determine $D_0$ and $b$, the diffusion coefficients obtained at different temperatures were fitted by nonlinear regression to

Eq. (5). They were weighted by the inverse of their statistical error where available. The results are summarized in Table 3 together with the diffusion coefficients calculated by the Chapman-Enskok theory using Eq. (1) and the Fuller method using Eq. (10). The input parameters used are summarized in Table 1.





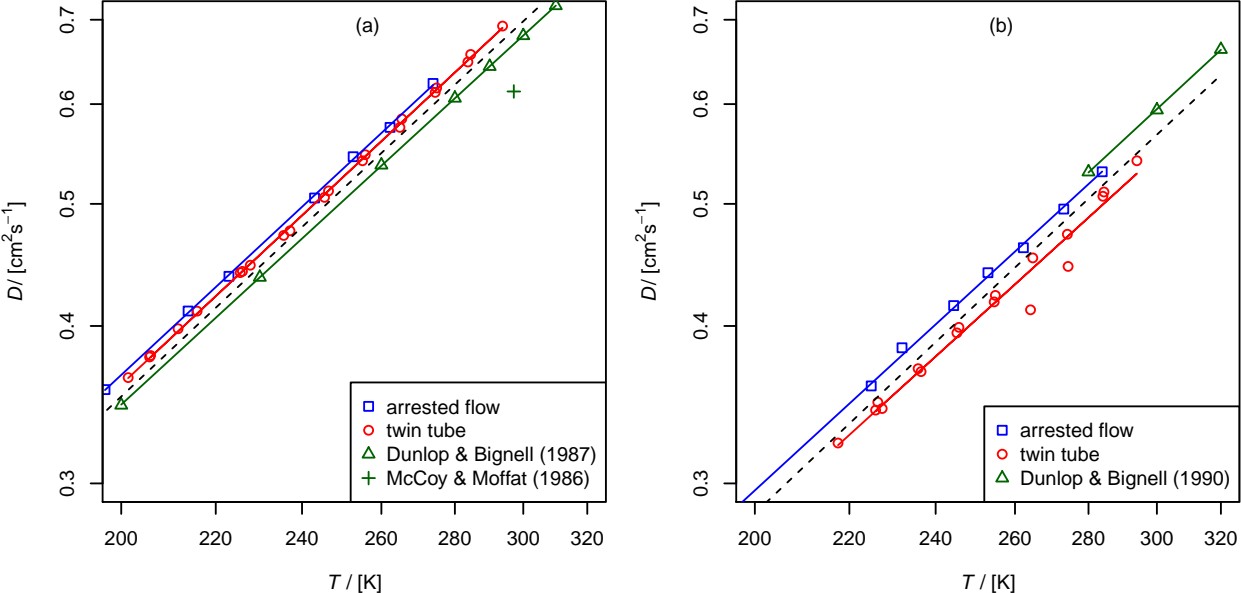

**Figure 6.** Comparison of diffusion coefficients obtained by the AF and TT method: (a) diffusion coefficient of methane in helium, compared to data of Dunlop and Bignell (1987) (fit $D_0 = 0.582 \, \mathrm{cm^2 s^{-1}}$, $b = 1.66$) and $D = 0.614 \, \mathrm{cm^2 s^{-1}}$ at 297 K of McCoy and Moffat (1986). (b) Diffusion coefficient of ethene in helium compared to data of Dunlop and Bignell (1990) (fit $D_0 = 0.508 \, \mathrm{cm^2 s^{-1}}$, $b = 1.68$). The dashed line is calculated by the Lennard-Jones model.

**Table 2.** Measured diffusion coefficients $D_0$ at the reference temperature $T$ and reference data of diffusion coefficients $D_r$. With 298 K as reference temperature, the measured diffusion coefficients were extrapolated to this temperature using Eq. (5).

| Species | Carrier | $T$ | $D_r$ | $D_0/D_r - 1$ | |
|---|---|---|---|---|---|
| | | [K] | [cm$^2$s$^{-1}$] | AF | TT |
| NO | He | 273 | $0.624_a$ | 6% | 7% |
| CH$_4$ | He | 273 | $0.582_b$ | 6% | 5% |
| C$_2$H$_4$ | He | 273 | $0.508_c$ | -2% | -8% |
| CH$_4$ | air | 298 | $0.221_d$ | 8% | 0% |
| C$_2$H$_4$ | air | 298 | $0.163_d$ | 8% | -4% |

Source: a Dunlop and Bignell (1992), b Dunlop and Bignell (1987), c Dunlop and Bignell (1990), d Tang et al. (2015)





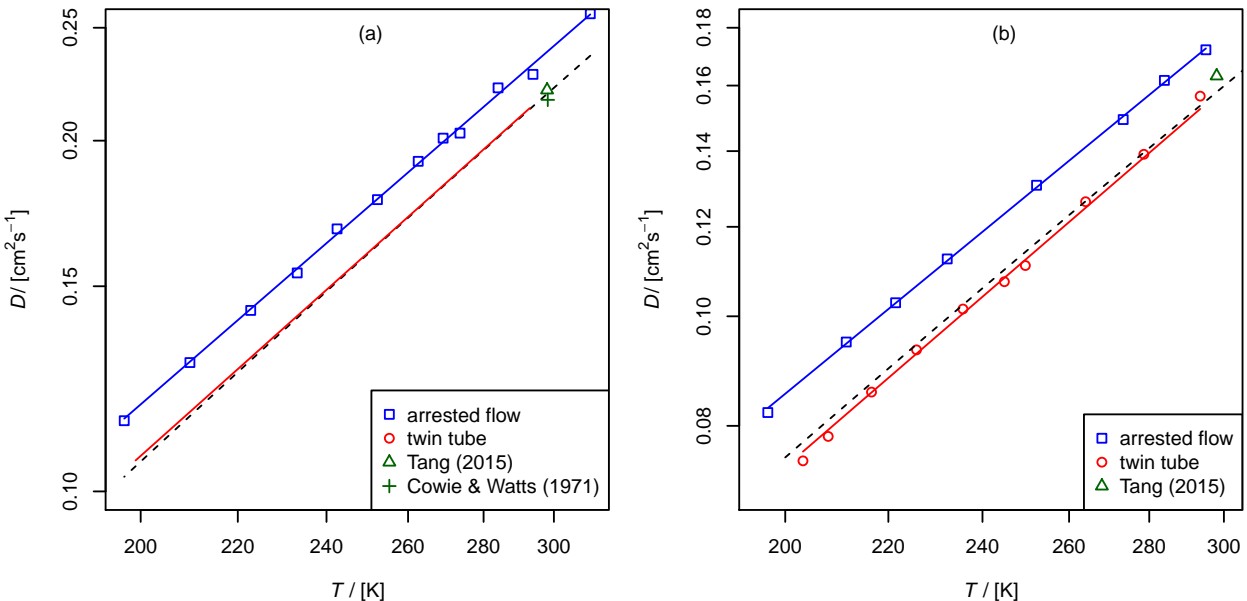

**Figure 7.** (a) Diffusion coefficient of methane in air, recommended value of $D = 0.221$ cm$^2$s$^{-1}$ at 298 K of Tang et al. (2015) and $D = 0.217$ cm$^2$s$^{-1}$ of Cowie and Watts (1971) at 298.2 K. For the TT method only the fit function is shown. (b) Diffusion coefficient of ethene in air as function of temperature, recommended value of Tang et al. (2015) $D = 0.163$ cm$^2$s$^{-1}$ at 298 K. The dashed line is calculated using the Lennard-Jones model.

## 3.1 Method validation

### 3.1.1 Diffusion of methane (CH$_4$) and ethene (C$_2$H$_4$) in helium and air

These gases were investigated with both methods for validation purposes. They are stable and non adsorbing, reference literature data of diffusion coefficients do exist. The diffusion coefficients in helium have been measured previously over a wider temperature range with high precision and accuracy by Dunlop and Bignell (1987) for methane and Dunlop and Bignell (1990) and ethene. Evaluated diffusion coefficients of hydrocarbons in air at 298 K are reported in the review of Tang et al. (2015). In addition, the diffusion coefficients can be calculated using the Lennard-Jones model and the Chapman-Enskok theory using 160 Eq. (1).

We have used a flame ionization detector (Carlo Erba FID 40 with EL980 control unit) which is fast, sensitive and linear over a wide concentration range to measure the hydrocarbons. For the AF experiments, 0.1% of methane or ethane in helium or air were injected as a 300 ms pulse. About 20–26 sccm were used as flow rate. The arrest time was varied from 0 to 200 s. For the TT experiments 0.5% and 1% methane in air, 0.4% methane in He, 0.4% ethene in He and 0.5% ethene in air were admitted in 165 the flow tube. Downstream of the diffusion bridge, the trace gas was analyzed using the peak mode setup, see Fig. 4.





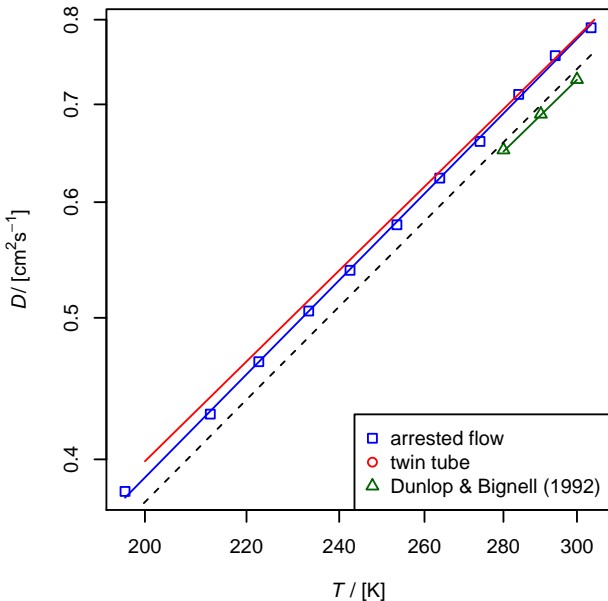

**Figure 8.** Diffusion coefficient of NO in He measured by AF and TT method as function of temperature. For the TT method only the fit function is shown. Experimental results of Dunlop and Bignell (1992) and a fit to Eq. (5) with $D_0 = 0.624$ and $b = 1.64$ are displayed as reference. The dashed line is calculated using the Lennard-Jones model.

The results are summarized in Table 2 and Table 3, in Fig. 6 and Fig. 7. Higher diffusion coefficients were found for the AF method compared to the TT-method.

### 3.1.2 Diffusion of nitric oxide (NO) in helium

NO was monitored by a chemiluminescence detector (Marić et al., 1989) which was adapted to the lower flow rates of the diffu-
sion experiments. In the detector, NO reacts with $O_3$ in a low pressure reaction chamber (0.9–2 mbar) in a chemiluminescence reaction:

$$NO + O_3 \rightarrow NO_2 + O_2 + h\nu \tag{R1}$$

The emitted photons were detected using a Hamamatsu R562 photomultiplier tube.

For the AF experiment 100 ppm NO were injected as 300 ms pulse into the flow tube with a flow rate of 22.5 sccm. The
valves were connected using stainless steel tubes. For the TT experiment 30–70 ppm NO in He were admitted into the diffusion bridge. The setup displayed in Fig 3 was used to monitor NO in the continuous mode. It was found that after measuring the high concentration $c_0$, it needs several hours until the baseline has stabilized when measuring clean carrier gas. Therefore, it is not possible to measure $c$ and $c_0$ in succession. Thus, first $c_0$ was measured at room temperature. Then the detector was switched to clean carrier gas until the baseline has stabilized. Then $c$ was measured after lowering the temperature until the





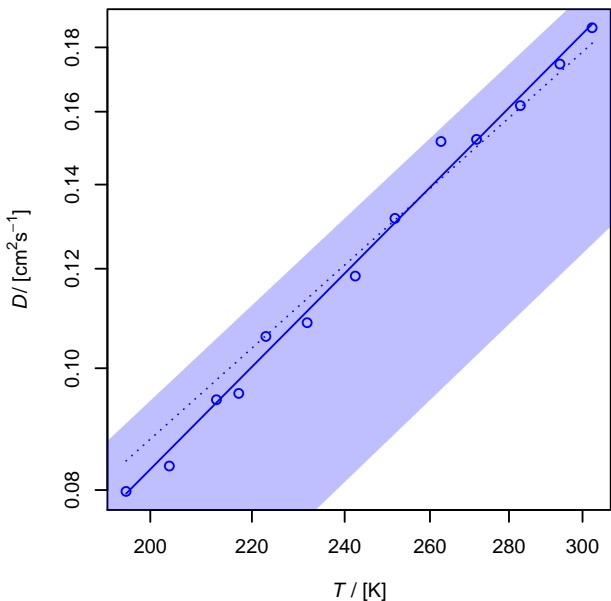

**Figure 9.** Diffusion coefficient of $O_3$ in air as function of temperature. The shaded area shows the diffusion coefficient expected by the Lennard-Jones model. The upper border is calculated with $D(T, 0.8\sigma_{AB}, 0.8\epsilon_{AB})$, the lower border with $D(T, 1.2\sigma_{AB}, 1.2\epsilon_{AB})$ using Eq. (1) corresponding to a 20% uncertainty of $\sigma_{AB}$ and $\epsilon_{AB}$. The dotted line is calculated by Fuller's model.

signal has stabilized. After arriving at 200 K, $c_0$ was measured again. Then the measurement was repeated by stepwise raising the temperature. Therefore, the complete measurement extended over several days.

The diffusion coefficients for NO obtained by the two methods are in fair agreement with the reference data of Dunlop and Bignell (1992), see Fig. 8 and Table 2. In contrast to diffusion coefficients of methane and ethane, the diffusion coefficients obtained by the TT method are slightly larger than the diffusion coefficients obtained by the AF method.

Comparing all diffusion coefficients obtained for stable gases to reference data, it is found that the deviation of the AF and TT method is less than 8% compared to the reference data. However, for the TT method this is a little more, as expected as theoretical systematic error, which can be explained by decreasing effective areas of the diffusion capillaries.

### 3.2 Diffusion of atmospheric trace gases

### 3.2.1 Diffusion of ozone ($O_3$) in air

The diffusion coefficient of ozone in air has never been measured before. Ivanov et al. (2007) reported $D = (0.53\pm0.03)\,\mathrm{cm^2 s^{-1}}$ for the diffusion of $O_3$ in He at 298 K. Since ozone is an unstable but non-adsorbing species, only the AF method was used for the determination of the diffusion coefficient. A fast and sensitive ozone detector is required to record the ozone peaks leaving the column. A suitable detection technique is chemiluminescence arising from the reaction of ozone with Coumarin





47 (Lambda Physik, 7-diethylamino-4-methylcoumarin), adsorbed on silica-gel plates (Schurath et al., 1991). Chemilumines-

cence is emitted in the range $\lambda = 440 - 550$ nm which is detected by photomultiplier (Hamamatsu 931 B). The anode current was admitted through a $100\,\mathrm{k}\Omega$ resistor and measured as voltage by a microvoltmeter (Keithley model 155).

Ozone containing air was generated in an aluminum block enclosing an elliptically shaped polished chamber. A rod-shaped low-pressure Hg UV-lamp and a quartz tube with air running through are mounted parallel in the focal lines of the elliptically shaped chamber. Thereby, the UV radiation is focused to the air flowing through the quartz tube (Becker et al., 1975). 5 blocks

arranged in series were used yielding an ozone concentration of about 40 ppm.

The injection time was varied from 250 ms to 500 ms and the arrest time was varied from 0 s to 360 s. It was found that the maxima of the eluted peaks did not coincide: peaks arrested longer were eluted later. Later it was found that this was caused by leaking sealings of the solenoid valves, which were made of neoprene. Therefore, at some temperatures only arrest times of less than 100 s were included in the fit of $\varsigma_z^2$ vs. $t_a$. The statistical error of the slope of $\varsigma_z^2$ vs. $t_a$ was up to 3.7%.

With regard to the systematic error of 7% of the AF method for the diffusion coefficients (Section 3.1.1), the obtained value with error ranges is $D_0 = 0.15 \pm 0.01\ \mathrm{cm^2 s^{-1}}$. This value is in accordance with the value of $D_0 = 0.1444\ \mathrm{cm^2 s^{-1}}$ estimated by Massman (1998) from critical constants using the model of Chen and Othmer (1962).

### 3.2.2  Diffusion of nitrogen dioxide ($NO_2$) and dinitrogen tetroxide ($N_2O_4$) in helium and nitrogen

$NO_2$ is in equilibrium with its dimer

$2\,NO_2 \rightleftharpoons N_2O_4$ (R2)

Therefore, a pure sample of $NO_2$ for determinations of $D$ is not available. Chambers and Sherwood (1937) assumed that the ratio of $D(NO_2)/D(N_2O_4) = 1.43$ yielding $D_0(NO_2) = 0.121\ \mathrm{cm^2 s^{-1}}$ in nitrogen from their value of $D_0(N_2O_4) = (0.0845 \pm 0.0005)\ \mathrm{cm^2 s^{-1}}$. Massman (1998) estimated $D_0(NO_2) = 0.146\ \mathrm{cm^2 s^{-1}}$ from $D_0(N_2O_4) = 0.101\ \mathrm{cm^2 s^{-1}}$ in nitrogen reported by Sviridenko et al. (1973). Since $NO_2$ is an adsorbing species, the diffusion coefficient can only be measured

using the TT method. The total flux of the pseudo species $N_{IV} = NO_2 + 2\,N_2O_4$ though the capillaries is given by

$$J(N_{IV}) = \frac{D(NO_2)c_0(NO_2) + 2D(N_2O_4)c_0(N_2O_4)}{l}.$$ (19)

At higher temperatures and when keeping the concentration of $NO_2$ low, diffusion of $N_2O_4$ can be neglected. The degree of dissociation

$$\alpha = \frac{p(NO_2)}{p(N_{IV})}$$ (20)

can be calculated from the equilibrium constant

$$K = \frac{p^2(NO_2)}{p^{\ominus} p(N_2O_4)} = \frac{p(N_{IV})}{p^{\ominus}} \frac{2\alpha^2}{1 - \alpha}$$ (21)

where $p^{\ominus} = 1$ bar. The equilibrium constant close to 250 K is estimated from JANAF Thermochemical Tables (NIST, 1998)

$$\ln K = 21.16 - 6878.1\,\mathrm{K}/T.$$ (22)





**Table 3.** Results of fit of Eq. (5) to the measured data. The errors listed are the errors obtained by the fit. Diffusion coefficients calculated by the Lennard-Jones and Fuller method are displayed for comparison. $D_0(N_2O_4)$ is estimated by nonlinear regression of Eq. (23) arbitrarily setting $b = 1.75$. For $N_2O_5$ the fit parameters of Eq. (24) are displayed.

| Species | Carrier | Method | | Experimental | | Lennard-Jones model | | Fuller et al. (1966) |
|---|---|---|---|---|---|---|---|---|
| | | | $T$ | $D_0$ | $b$ | $D_0$ | $b$ | $D_0$ |
| | | | [K] | [cm$^2$s$^{-1}$] | | [cm$^2$s$^{-1}$] | | [cm$^2$s$^{-1}$] |
| CH$_4$ | He | AF | 197–274 | 0.618±0.002 | 1.68±0.02 | 0.596 | 1.68 | 0.549 |
| CH$_4$ | He | TT | 201–294 | 0.610±0.001 | 1.69±0.01 | " | " | " |
| CH$_4$ | air | AF | 197–311 | 0.205±0.001 | 1.75±0.02 | 0.188 | 1.80 | 0.180 |
| CH$_4$ | air | TT | 199–293 | 0.188±0.003 | 1.80±0.04 | " | " | " |
| C$_2$H$_4$ | He | AF | 197–284 | 0.497±0.002 | 1.66±0.03 | 0.484 | 1.70 | 0.418 |
| C$_2$H$_4$ | He | TT | 218–294 | 0.468±0.004 | 1.65±0.08 | " | " | " |
| C$_2$H$_4$ | air | AF | 197–295 | 0.150±0.000 | 1.81±0.01 | 0.135 | 1.84 | 0.128 |
| C$_2$H$_4$ | air | TT | 203–294 | 0.133±0.001 | 1.90±0.05 | " | " | " |
| NO | He | AF | 196–304 | 0.662±0.001 | 1.71±0.02 | 0.633 | 1.68 | 0.757 |
| NO | He | TT | 200–305 | 0.667±0.013 | 1.65±0.02 | " | " | " |
| NO$_2$ | He | TT | 251–299 | 0.520±0.001 | 1.93±0.03 | 0.537 | 1.70 | 0.613 |
| NO$_2$ | N$_2$ | TT | 251–319 | 0.145±0.001 | 1.94±0.06 | 0.135 | 1.82 | 0.162 |
| O$_3$ | air | AF | 196–303 | 0.153±0.001 | 1.97±0.06 | 0.131 | 1.83 | 0.152 |
| N$_2$O$_4$ | He | TT | 202–299 | 0.221±0.014 | 1.75±0.25 | 0.388 | 1.71 | 0.440 |
| N$_2$O$_4$ | N$_2$ | TT | 204–319 | 0.084±0.004 | 1.75±0.25 | 0.090 | 1.88 | 0.115 |
| ClONO$_2$ | He | AF | 236–304 | 0.310±0.004 | 1.57±0.17 | 0.402 | 1.72 | 0.387 |
| ClONO$_2$ | N$_2$ | AF | 234–298 | 0.085±0.001 | 2.27±0.21 | 0.092 | 1.88 | 0.103 |
| N$_2$O$_5$ | He | AF | 245–298 | 0.300±0.012 | 1.39±0.23 | 0.381 | 1.73 | 0.405 |
| N$_2$O$_5$ | N$_2$ | AF | 246–298 | 0.081±0.005 | 1.66±0.26 | 0.085 | 1.91 | 0.106 |





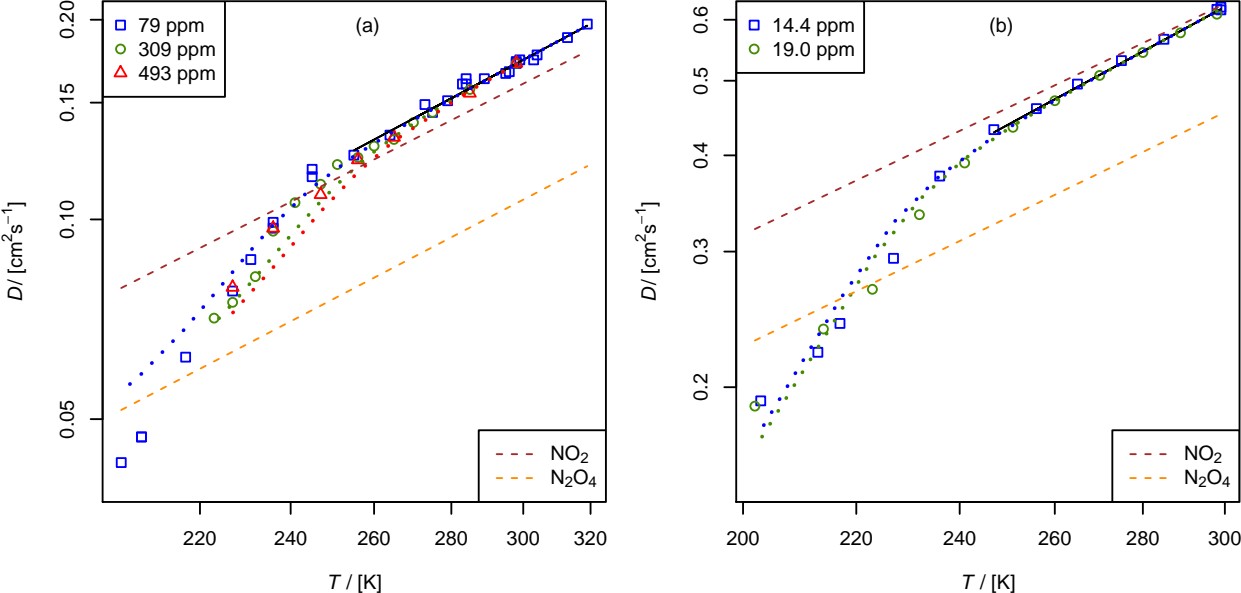

**Figure 10.** Apparent diffusion coefficient $D(\mathrm{N_{IV}})$ of $\mathrm{NO_2}/\mathrm{N_2O_4}$ in $\mathrm{N_2}$ (a) and He (b) as function of temperature. Black solid line: fit of Eq. (5) to data points with $\alpha > 0.95$. The dashed lines are calculated using the Lennard-Jones model. The dotted curves are fit curves of Eq. (23) by setting $b(\mathrm{N_2O_4}) = 1.75$.

Actually $D(\mathrm{N_{IV}})$ is determined using the TT experiment, depending on $D(\mathrm{NO_2})$ and $D(\mathrm{N_2O_4})$ (see Supplement S1):

$$D(\mathrm{N_{IV}}) = \alpha D(\mathrm{NO_2}) + (1 - \alpha) D(\mathrm{N_2O_4}) \tag{23}$$

A thermostatted permeation tube consisting of a PTFE tube (2.5 cm length, 4.3 mm diameter) closed with two Swagelok connectors filled with liquid $\mathrm{N_2O_4}$ is used as $\mathrm{NO_2}$-source. The measurements were performed in the temperature range from 200 K to 300 K. $\mathrm{NO_2}$ was measured as NO using the chemiluminescence analyzer preceded by a thermal converter. The converter, which consisted of a gold wire in a thin quartz tube heated to 540 K, was run with 1.5% – 3% methanol vapor instead of the more commonly used CO reagent (Langenberg et al., 1998). This eliminated poisoning of the gold wire which occurs when metal carbonyl impurities are present in the CO gas. The total conversion rate is unknown, However, for the TT experiment, it is only required that the conversion rate is independent of concentration, which was checked for the concentration range of 3 to 100 ppm $\mathrm{NO_2}$ by a dynamic dilution experiment. $\mathrm{NO_2}$ was monitored using the continuous mode of Fig. 5. In order to compensate for aging effects of the capillary bridge, $\mathrm{CH_4}$ was admixed as an internal standard to the carrier gas flowing through the $\mathrm{NO_2}$ permeation source. The 4-port valve in front of the detector was replaced by a 6-port valve (Valco UC10W, 125 $\mu$m sample loop) which enables detecting $\mathrm{CH_4}$ by the flame ionization detector. For $\mathrm{CH_4}$ in He $D_0 = (0.582 \pm 0.003)$ cm$^2$s$^{-1}$ at 273.15 K (Dunlop and Bignell, 1987) and for $\mathrm{CH_4}$ in $\mathrm{N_2}$ $D = (0.216 \pm 0.001)$ cm$^2$s$^{-1}$ at 298 K (Mueller and Cahill, 1964) was chosen as reference diffusion coefficient for the internal standard. The measurement was performed in





a similar manner like the NO measurement with the TT experiment by stepwise lowering and rising the temperature. Above
250 K the statistical error of the low concentration is about 3%. However, below 250 K signal stability was low. In contrast
to the measurements above 250 K, apparent $D(\mathrm{N_{IV}})$ measured at a certain temperature by stepwise lowering the temperature
does not reproduce the value of $D(\mathrm{N_{IV}})$ measured by stepwise rising the temperature.

Figure 10 displays the obtained diffusion coefficients of the pseudo species $\mathrm{N_{IV}}$ as function of temperature. It is obvious
that below 250 K the plot is deviating. Above 250 K diffusion of $\mathrm{N_2O_4}$ can be neglected because $\mathrm{N_2O_4}$ is mostly dissociated
in the concentration range of our study. To estimate the diffusion coefficient of $\mathrm{NO_2}$ only data points with dissociation degree
$\alpha > 0.95$ were included in the fit of Eq. (5). Thus, regarding the errors of the diffusion coefficients of the internal standards,
the diffusion coefficients for $\mathrm{NO_2}$ at STP are $D_0 = (0.520 \pm 0.004)\ \mathrm{cm^2 s^{-1}}$ and $D_0 = (0.145 \pm 0.002)\ \mathrm{cm^2 s^{-1}}$ in helium and
nitrogen, respectively.

To determine $D_0(\mathrm{N_2O_4})$, Eq. (23) was fitted to experimental data of $D(\mathrm{N_{IV}})$ vs. $T$ and $p(\mathrm{N_{IV}})$ by nonlinear regression.
The temperature dependency of $D(\mathrm{N_{IV}})$ was described by Eq. (5) and $\alpha$ as function of temperature and $p(\mathrm{N_{IV}})$ was calculated
using Eq. (21). $D_0(\mathrm{NO_2})$ and $b(\mathrm{NO_2})$ were taken as fixed input parameters from the fit of Eq. (5) as described above. However,
an independent determination of $D_0(\mathrm{N_2O_4})$ and $b(\mathrm{N_2O_4})$ was not possible. Therefore, $b(\mathrm{N_2O_4}) = 1.75$ was set arbitrarily,
yielding the diffusion coefficients listed in Table 3. Since $b(\mathrm{N_2O_4})$ is expected in the range 1.5 to 2, the fit was repeated setting
$b = 2$ and $b(\mathrm{N_2O_4}) = 1.5$ to estimate the upper and lower limit of $D_0(\mathrm{N_2O_4})$ listed in Table 3. Compared to the diffusion
coefficient of $\mathrm{N_2O_4}$ in $\mathrm{N_2}$, the diffusion coefficient in He determined by the fit is much lower than the diffusion coefficient
calculated by the Lennard-Jones model. In addition, our values are lower than the values of Sviridenko et al. (1973). We
therefore consider our diffusion coefficients for $\mathrm{N_2O_4}$ to be unreliable. However, we can explain the observed temperature
dependency of $D(\mathrm{N_{IV}})$ in the transition to lower temperatures.

### 3.2.3 Diffusion of chlorine nitrate ($\mathrm{ClONO_2}$) in helium and nitrogen

Chlorine nitrate is an unstable compound. Therefore, the diffusion coefficient can only be measured using the AF-method and
not by the TT-method. Chlorine nitrate was prepared by the reaction (Davidson et al., 1987):

$$\mathrm{Cl_2O + N_2O_5 \rightarrow 2\,ClONO_2} \tag{R3}$$

$\mathrm{Cl_2O}$ was prepared by admitting chlorine into a column filled with Raschig rings covered with freshly precipitated HgO
(Schmeisser et al., 1967):

$$\mathrm{2\,Cl_2 + HgO \rightarrow Cl_2O + HgCl_2} \tag{R4}$$

The formed $\mathrm{Cl_2O}$ was condensed over $\mathrm{N_2O_5}$ in a cold trap cooled with liquid nitrogen. Then the cold trap was cooled with
Ethanol at 193 K which was allowed to warm up to 253 K within about 15 h.

The identity of the product was checked by a FTIR spectrometer (Nicolet, model Protégé 460) with 10 m absorption path.
The spectrum was recorded with $1\ \mathrm{cm^{-1}}$ resolution. $\mathrm{ClONO_2}$ was characterized by typical bands at 535–580 $\mathrm{cm^{-1}}$, 750–
825 $\mathrm{cm^{-1}}$, 1270–1320 $\mathrm{cm^{-1}}$ and 1695–1770 $\mathrm{cm^{-1}}$ compared to reference spectra measured by Davidson et al. (1987) and
Orphal et al. (1997). No contamination of $\mathrm{N_2O_5}$, $\mathrm{NO_2}$ and HCl was found.





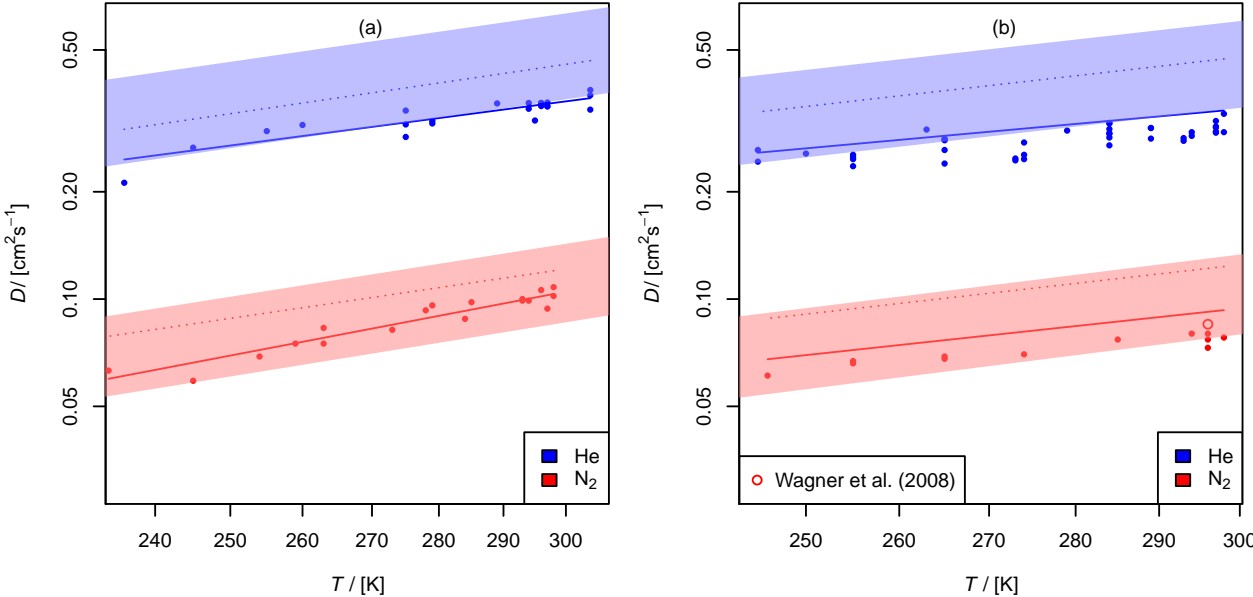

**Figure 11.** (a) Diffusion coefficient of chlorine nitrate in helium and nitrogen, (b) diffusion coefficient of dinitrogen pentoxide in helium and nitrogen as function of temperature The experimental result of $0.085\ \mathrm{cm^2 s^{-1}}$ of Wagner et al. (2008) is displayed for comparison. The shaded areas show the diffusion coefficients expected by the Lennard-Jones model. The dotted lines are calculated by Fuller's model. The solid lines displayed for $N_2O_5$ are calculated using Eq. (24) extrapolating to $k_1 = 0\ \mathrm{s^{-1}}$.

Chlorine nitrate dissociates by the equilibrium reaction:

$$ClONO_2 + M \rightleftharpoons ClO + NO_2 + M. \tag{R5}$$

The half-life of chlorine nitrate with respect to the thermal decomposition is 11 min at 300 K and about 7.8 h at 273 K. In
addition to homogeneous dissociation, chlorine nitrate is lost by heterogeneous reaction with adsorbed water on the column surface (Tang et al., 2016):

$$ClONO_2 + H_2O \rightarrow HOCl + HNO_3 \tag{R6}$$

To minimize this interference, the column was preconditioned with chlorine nitrate to remove moisture. During the first series of measurements with He as carrier gas, chlorine nitrate was continuously admitted into the column for 10–15 min prior to
the experiments. However, due to desorption the base line stabilized only after some time. Therefore, during the second series of measurements with $N_2$ as carrier gas, series of peaks of chlorine nitrate were admitted into the column, until the peak size had stabilized. During one series of measurements 16 peaks covering arrest times from 0 to 100 s and a carrier gas flow rate of 19.6 sccm $N_2$ and 28.6 sccm He were measured. The diffusion coefficient was measured in a temperature range of 235 K to 300 K.





The detection of chlorine nitrate was performed as described by Anderson and Fahey (1990): an excess of NO (30–75 ppm) was added as constant flow of 6.5–8 sccm in $N_2$ before the detector by a T-tube. Behind the T-tube a glass capillary of 8 cm length and 0.1 cm inner diameter was mounted. The capillary was inserted in a stainless steel tube which was heated on a length of 1.7 cm to 433 K by two heating resistors. In the heating zone, chlorine nitrate is dissociated to ClO by Reaction R5. By subsequent scavenging reactions, NO is irreversibly removed:

$ClO + NO \rightarrow Cl + NO_2$                                                                (R7)

$Cl + ClONO_2 \rightarrow Cl_2 + NO_3$                                                      (R8)

$NO_3 + NO \rightarrow 2\,NO_2$                                                              (R9)

A complete conversion of chlorine nitrate with NO is assumed. The drop of NO concentration equals the $ClONO_2$ concentration and was monitored using the chemiluminescence detector.

It is presumed that the chlorine nitrate loss processes during peak arrest is a pure first order process, which is a requirement for the application of the AF-method. To check the first order kinetics, logarithmic peak areas as measure for the chlorine nitrate concentration were plotted against the arrest time yielding a straight line. This validated the first order characteristic of

the chlorine loss process. The first order loss constants ranged from $8.8 \times 10^{-4}\ \mathrm{s}^{-1}$ to $4.9 \times 10^{-3}\ \mathrm{s}^{-1}$. During the experiments with $N_2$ as carrier gas, loss rates increased with decreasing temperature. During the experiments with He, loss rates increased with increasing temperature. However, for the He experiments, a less effective preconditioning technique was applied, as described above.

The diffusion coefficients obtained at different temperatures are displayed in Fig.11 (a). When taking a systematic error

of 7% for the AF method into account, the diffusion coefficients at STP are $D_0 = (0.31 \pm 0.03)\ \mathrm{cm^2 s^{-1}}$ and $D_0 = (0.085 \pm 0.007)\ \mathrm{cm^2 s^{-1}}$ in helium and nitrogen, respectively.

### 3.2.4   Diffusion of dinitrogen pentoxide ($N_2O_5$) in helium and nitrogen

Crystalline $N_2O_5$ was synthesized as described by Davidson et al. (1978) and Tang et al. (2014b): a small flow of pure NO is mixed with $O_3/O_2$ in a glass reactor and trapping the product at 193 K using a cold trap immersed in a cold ethanol bath. $O_3$

was generated in pure dry $O_2$ with a silent discharge ozone generator (Sorbios, model GSG). After mixing NO with $O_3$ / $O_2$ in the glass reactor, brown color appeared initially, indicating the formation of $NO_2$:

$NO + O_3 \rightarrow NO_2 + O_2,$                                                          (R10)

$NO_2 + O_3 \rightarrow NO_3 + O_2,$                                                        (R11)






$$NO_2 + NO_3 + M \rightarrow N_2O_5 + M. \qquad (R12)$$

After about 3 h, addition of NO is stopped and the cooling bath of the cold trap is removed. To remove traces of $NO_2$, the product is transferred into a second cold trap using a $O_3$ / $O_2$ stream. By means of a cryostat, the synthesized $N_2O_5$ crystals were stored in an ethanol bath kept at 193 K. The identity of the product was checked by infrared spectroscopy. The spectrum

was recorded with 1 $cm^{-1}$ resolution. $N_2O_5$ was characterized by typical bands at 750, 860, 1250, 1340 and 1725 $cm^{-1}$ reported by a reference spectrum measured by Cantrell et al. (1988).

The cold trap filled with $N_2O_5$ crystals was immersed in the bath of a cryostat (Lauda RLS 6) thermostatted at 235 K to 250 K. Dried carrier gas was admitted through the cold trap. An upper limit of the partial pressure of $N_2O_5$ in contact with the solid can be estimated from data of McDaniel et al. (1988). This results in an upper limit of the mole fraction of about 0.1% to

0.6%. The trace gas was admitted into the AF-experiments using short Teflon tubes.

$N_2O_5$ was detected as already described by Fahey et al. (1985): $N_2O_5$ is thermally decomposed to $NO_2$ and $NO_3$ radicals which are then titrated by NO to form $NO_2$

$$N_2O_5 + M \rightarrow NO_2 + NO_3 + M, \qquad (R13)$$

$$NO_3 + NO \rightarrow 2NO_2. \qquad (R14)$$

The drop in NO concentration is equal to the $N_2O_5$ concentration. NO was measured again by the chemiluminescence detector. Downstream of the AF-experiment, a constant flow of 6 – 8 sccm of 30 – 45 ppm NO was added.

The diffusion coefficient of $N_2O_5$ was measured in the temperature range 245 K to 298 K. Below 245 K no measurement was possible because $N_2O_5$ was totally adsorbed in the column. At one fixed temperature 16 peaks were measured using

arrest times between 0 s and 100 s. At the beginning of a measurement series, the column was preconditioned with carrier gas containing $N_2O_5$ to remove moisture for 15 – 20 min. Prior of a measurement with arrest time, a peak without arrest time was pushed trough the column. During one series of measurements 16 peaks covering arrest times from 0 to 100 s and a carrier gas flow rate of 19.3 sccm $N_2$ and 28.8 sccm He were measured.

Equation (5) was used to obtain $D_0 = (0.276 \pm 0.003)$ $cm^2 s^{-1}$, $b = (1.0 \pm 0.2)$ in He and $D_0 = (0.0709 \pm 0.0006)$ $cm^2 s^{-1}$,

$b = (1.1 \pm 0.1)$ in $N_2$. Thus, the observed temperature coefficient is much lower than expected from Chapman-Enskok theory. As long as the $N_2O_5$ loss process is purely first order, the peak variance should not be affected by loss processes. To check this, peak areas were determined by integration. Plots of log (peak area) versus arrest times yielded apparent first order loss constants $k_1$, ranging from $4 \times 10^{-3}$ $s^{-1}$ to $2 \times 10^{-2}$ $s^{-1}$. To check if $D$ depends on $k_1$, the diffusion coefficient was expressed by

$$D = D_0 \left( \frac{p_0}{p} \right) \left( \frac{T}{T_0} \right)^b \exp(ak_1) \qquad (24)$$



as function of $T$ and $k_1$. It was found that $D$ not only significantly depends on $T$, but also on $k_1$. With $P$, the probability of the null hypothesis, it was found for He and N$_2$ $a = -(8 \pm 4)$ s ($P < 0.05$) and $a = -(21 \pm 9)$ s ($P < 0.06$), respectively. One reason for this may be that the order of the loss process of N$_2$O$_5$ is less than first order. The other fit parameters are displayed in Table 3. For He as carrier gas, a temperature coefficient of $b < 1.5$ was found. Due to the small temperature range investigated, the temperature coefficient is rather uncertain. As final result, for He $D_0 = (0.30^{+0.03}_{-0.06})$ cm$^2$s$^{-1}$ and N$_2$ $D_0 = (0.08^{+0.01}_{-0.02})$ cm$^2$s$^{-1}$ are obtained when considering the systematic error of 7% for the AF method and the possible influence of dinitrogen pentoxide degradation on the diffusion coefficients obtained. Wagner et al. (2008) reported a diffusion coefficient of 0.085 cm$^2$s$^{-1}$ for N$_2$O$_5$ in N$_2$ at 760 Torr and 296 K, which is within the error limits of our result.

## 4 Conclusions

The AF method is best suited for the measurement of diffusion coefficients of volatile non adsorbing trace gases, even if the trace gas is unstable like ozone. However, it is required that the trace gas loss process is first order. Otherwise, the Gaussian peak shape is distorted and the variance of the peaks depends on species reactions. The TT method is best suited for stable but adsorbing species.

For stable non polar gases diffusion coefficients can be estimated from viscosity data using the Lennard-Jones model with a systematic error of $< 5\%$, which is smaller than the systematic errors of less than 7% of the AF and TT-methods. For unstable atmospherically relevant trace gases and polar gases, the Lennard-Jones model parameters cannot be obtained by viscosity measurements. They can only be estimated from critical temperatures and volumes. Where dipole – induced dipole interactions come into play the systematic errors of the diffusion coefficients obtained in this way are in the same order as the errors of the diffusion coefficients of the unstable and reactive trace gases investigated in this study.

For the species investigated in this study, it is found that Fuller's method overestimates diffusion coefficients of inorganic compounds with a systematic error of typically less than 35% and underestimates diffusion coefficients of organic compounds with a systematic error of less than 15%.

*Code and data availability.* Raw data of temperature dependent diffusion coefficients are included in the supplement. Sample code for the calculation of diffusion coefficients using the Lennard-Jones and the Fuller model written in the language R (R Core Team, 2017) is also included.

*Author contributions.* Conceptualization and methodology (SS), investigation (TC, DH, SS), formal analysis and visualization (TC, DH, SL, SS), writing draft (SL), supervision and funding acquisition (US). All the authors have read and approved the final manuscript.

*Competing interests.* The authors declare that they have no conflict of interest.





*Acknowledgements.* This work was supported by the "Deutsche Forschungsgemeinschaft (DFG)" within the DFG priority program "Basics
of the Impact of Air and Space Transportation on the Atmosphere". TC acknowledges a doctoral grant from Karlsruher Institut für Technologie. We thank Ralf Rubröder and Birgit Walter for setting up the AF experiment; Peter Boecker for the assistance measuring FT-IR spectra; Harald Saathoff for providing infrared spectra of $N_2O_5$, HCl and $NO_2$; Ewald Hild for the preparation of electron micrographs of the fused silica columns and Dieter Gauer for technical assistance.



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
