# Peer review of "Technical note: Determination of binary gas phase diffusion coefficients of unstable and adsorbing atmospheric trace gases at low temperature – Arrested Flow and Twin Tube method"

_Atmospheric Chemistry and Physics, 2019_

## Short Comment (SC1) · 11 Dec 2019

As the leading author of a few very relevant papers (Tang et al., 2014a; Tang et al., 2015; Gu et al., 2018) cited by this manuscript, I would like to congratulate the authors on their nice work. I also have a few comments:

1) As diffusion coefficients depend on temperature, it took me a while to know the corresponding temperature for D0 presented in Tables 2-3. Although it is clear in the

text, it may be helpful to explain D0 in the table captions.

2) It will be useful to provide an outlook to tell the readers which trac gases will be (are being) investigated using the two nice techniques?

Dr. Mingjin Tang

Guangzhou Institute of Geochemistry

Chinese Academy of Sciences

Guangzhou 510640, China

---

## Author Comment (AC1) · 30 Dec 2019

We thank Mingjin Tang for commenting our discussion paper. His remarks are marked like *this*. All references, cited herein refer to the discussion paper. To the questions we answer as follows:

*1) As diffusion coefficients depend on temperature, it took me a while to know the*

[Figure]

*corresponding temperature for $D_0$ presented in Tables 2-3. Although it is clear in the text, it may be helpful to explain $D_0$ in the table captions.*

This will be done in the revised manuscript.

*2) It will be useful to provide an outlook to tell the readers which trace gases will be (are being) investigated using the two nice techniques?*

The experiments were performed 1992-1997. Encouraged by the reviews of Tang et al. (2014a, 2015), we decided to publish our data. Today, the experimental setups do not exist anymore. Besides of the diffusion coefficients reported in our paper, diffusion coefficients of $O_2$, $CCl_4$ and $CCl_2F_2$ in $N_2$ were investigated using the AF method. With $CH_4$ as internal standard, ethene; ethane; propene in air and propane; *n*-butane; isoprene in $N_2$ were measured simultaneously using the TT method.

---

## Referee Comment (RC1) · Anonymous Referee #1 · 14 Jan 2020

This manuscript describes the development and application of two techniques for the measurement of gas phase diffusion coefficients for reactive or sticky trace gases of atmospheric relevance. The two techniques had been carefully designed, the measurements have been carefully performed and thoroughly analyzed, with an eye on all conceivable pitfalls and uncertainties. The manuscript is very well organized and well written. I have only two rather minor and general comments. The effort of the authors to publish this old but very valuable material is highly appreciated; it will be an important

contribution to the atmospheric science community.

In the experimental part, the authors shortly mention the limiting cases for each of the two methods. How many wall collisions do the diffusing molecules typically undergo? Since either reversible adsorption or chemical reaction are affecting the transport kinetics, the authors could elaborate the limiting first order loss rate coefficient and the limiting residence time on the surface to lead to a noticeable impact on the analysis of the detector signals for each method. Could the method in turn be used to measure the surface residence time of sticky but non-reactive molecules through their effective diffusivity?

line 190: no need to decide whether ozone is adsorbing or non-adsorbing. Each molecule may adsorb. I suggest to simply mention chemical decay. Whether it undergoes reversible adsorption in addition seems not obvious (cf also previous comment) from the data but cannot be excluded.

---

## Author Comment (AC2) · 22 Jan 2020

We thank the referee for reviewing and commenting our discussion paper. The remarks of the reviewer are marked like *this*. All symbols and equations used and cited herein refer to the discussion paper unless otherwise indicated. To the comments and questions we answer as follows:

[Figure]

*In the experimental part, the authors shortly mention the limiting cases for each of the two methods. How many wall collisions do the diffusing molecules typically undergo?*

The collision rate per unit area is given by

$$Z_w = \frac{1}{4}c\bar{v}$$

where

$$\bar{v} = \sqrt{\frac{8kT}{\pi m}} = \sqrt{\frac{8RT}{\pi M}}$$

is the mean molecular velocity of the species under investigation. If the trace gas contact time $\tau$ is given, the (dimensionless) number of hits of an individual species molecule on the cylindrical surface can be estimated

$$N_{\text{hit}} = \frac{\tau\bar{v}}{2r}.$$

For the AF-method $\tau$ corresponds to the arrest time $t_a$. For the TT-method $\tau$ is the mean travel time of a molecule along the length $l$ of the capillary. This yields

$$\tau = \frac{l^2}{2D}.$$

For ozone at 273.15 K $\tau = 14.4$ s is found for the TT-method. Thus, the number of hits is $9 \times 10^6$ for the AF-method at a arrest time of 100 s and $6 \times 10^7$ for the TT-method.

*Since either reversible adsorption or chemical reaction are affecting the transport kinetics, the authors could elaborate the limiting first order loss rate coefficient and the limiting residence time on the surface to lead to a noticeable impact on the analysis of the detector signals for each method.*

The AF-method is not affected by first order loss. However, most heterogeneous loss processes are not strictly first order. The TT-method *is* affected by lost processes. A

tolerable upper limit for the uptake/reaction coefficient $\gamma$ for enabling the TT-method is derived as follows: lets assume that the ratio between the reactive flow $J_R$ to the surface and the diffusion flow through the capillary should not exceed 0.01. Thus, $J_R < 0.01 J_D$. The reactive flow the surface is given by

$$J_R = \frac{\bar{v}\gamma}{4l} \int_0^l \tilde{c}(z)\mathrm{d}z,$$

considering that the concentration $\tilde{c}$ is a linear function of the coordinate $z$ along the capillary tube. We obtain

$$\tilde{c}(z) = c_0 + \frac{z}{l}(c - c_0).$$

This yields

$$J_R = \frac{1}{8}\bar{v}\gamma(c_0 + c).$$

Thus, it follows with Eq. (16)

$$\gamma < 0.08 \frac{D}{\bar{v}l}\left(\frac{c_0 - c}{c_0 + c}\right) \approx 0.08 \frac{D}{\bar{v}l}.$$

For ozone at 273.15 K with $\bar{v} = 347.1$ m s$^{-1}$ $\gamma \ll 1.7 \times 10^{-7}$ is required for the successful application of the TT-method. Ozone destruction on similar quartz surfaces was already studied by Langenberg and Schurath (1999): $\gamma = 4.4 \times 10^{-7}$ was found for ozone at room temperature. Therefore, the TT-method is not suitable to determine the diffusion coefficient of ozone. In addition, further loss of ozone in other parts of the apparatus needs to be considered too.

*Could the method in turn be used to measure the surface residence time of sticky but non-reactive molecules through their effective diffusivity?*

The TT-method is not affected by non-reactive adsorption. However, the AF-method may be affected. The most simplistic model for adsorption is Henry's adsorption

isotherm: the surface concentration $q$ as function of the trace gas partial pressure $p$ is given by (Langenberg and Schurath, 2018)

$$q = K_H p.$$

The capacity ratio is defined as

$$k' = \frac{2RT}{r} K_H.$$

In case of adsorption it follows from the mass balance in the diffusion capillary

$$\left(\frac{\partial c}{\partial t}\right)_z = \frac{D}{1 + k'} \left(\frac{\partial^2 c}{\partial z^2}\right)_t.$$

Therefore, the diffusion coefficient measured by the AF-method would be smaller than the actual diffusion coefficient. Thus, if the real diffusion coefficient $D$ is known, measurement of the effective diffusion coefficients can be used to determine $k'$ and $K_H$. However, it is much easier to determine $k'$ using standard gas chromatography by measuring arrest times (first central moment) instead of measurement of peak broadening (second central moment).

*Line 190: no need to decide whether ozone is adsorbing or non-adsorbing. Each molecule may adsorb. I suggest to simply mention chemical decay. Whether it undergoes reversible adsorption in addition seems not obvious (cf also previous comment) from the data but cannot be excluded.*

We also tried to investigate the diffusion of $NO_2$ using the AF-method. However, adsorption gave rise to strong peak tailing which invalidated the AF-method.

**References**

Langenberg, S. and Schurath, U.: Ozone Destruction on Ice, Geophys. Res. Lett., 26, 1695–1698, https://doi.org/10.1029/1999GL900325, 1999.

Langenberg, S. and Schurath, U.: Gas chromatography using ice-coated fused silica columns: study of adsorption of sulfur dioxide on water ice, Atmos. Chem. Phys., 18, 7527–7537, https://doi.org/10.5194/acp-18-7527-2018, 2018.

---

## Referee Comment (RC2) · Anonymous Referee #2 · 5 Feb 2020

An impressive set of measurements are described in this paper. The authors have thoroughly presented their diffusion coefficient measurements and the theory behind them. The previous work in this area is adequately presented.

Something in the big picture is missing: the application to the laboratory kinetics measurements whose results might depend on the accuracy of the diffusion coefficients (the stated reason for this detailed work, lines 60-62.) How will the results of these new

measurement capabilities affect previously measured uptake coefficients? It seems that the ClONO2 and N2O5 diffusion coefficients might have the most impact in this regard. A recommended set of L-J parameters for these two species would be most interesting.

The concerns about the measurements center around these two molecules and the temperature dependencies of the D's. Both N2O5's and ClONO2's measured T-dependencies differ significantly from that expected for L-J interactions. While losses were addressed, it seems these anomalous T-dependencies suggest there is more to the story. The indirect detection method for these two species is worth some consideration. Hard to come up with reasons why these two molecules interacting with He and N2 should not be describable by L-J potentials.

---

## Author Comment (AC3) · 18 Feb 2020

We thank the referee for reviewing and commenting our discussion paper. The remarks of the reviewer are marked like *this*. All symbols, equations and references used and cited herein refer to the discussion paper unless otherwise indicated. To the comments and questions we answer as follows:

[Figure]

*Something in the big picture is missing: the application to the laboratory kinetics measurements whose results might depend on the accuracy of the diffusion coefficients (the stated reason for this detailed work, lines 60-62.) How will the results of these new measurement capabilities affect previously measured uptake coefficients? It seems that the $ClONO_2$ and $N_2O_5$ diffusion coefficients might have the most impact in this regard. A recommended set of L-J parameters for these two species would be most interesting.*

Most uptake experiments with $ClONO_2$ and $N_2O_5$ into liquid surfaces were performed by the droplet train technique. To determine the real uptake coefficient $\gamma_0$, the measured uptake coefficient $\gamma_{obs}$ had to be corrected for diffusion to the droplet surface. This can be performed by a simplistic resistance model of Hu et al. (1995)

$$\frac{1}{\gamma} = \frac{1}{\gamma_{obs}} - \frac{\bar{v}r}{4D_g} + \frac{1}{2}$$

where $\bar{v}$ is the average trace gas thermal velocity and $r$ the particle radius. The necessary diffusion coefficients were mostly taken from Hanson and Ravishankara (1991), who estimated them using the Lennard-Jones model and parameters of Patrick and Golden (1983) listed in our Table 1. In general, the investigators did not perform a sensitivity analysis of the dependency of the diffusion coefficient upon their reported $\gamma$. Therefore, it is difficult to assess, how a change of the diffusion coefficient would change the value of $\gamma$ obtained.

We try to estimate this interdependency using the work of George et. al. (1994), who studied the uptake of $N_2O_5$ into water droplets. The dependency of the relative error of $\gamma$ on the relative error of the diffusion coefficient $D_g$ is given by

$$\frac{\Delta\gamma}{\gamma} = \frac{D_g}{\gamma}\left(\frac{\partial\gamma}{\partial D_g}\right)\frac{\Delta D_g}{D_g}.$$

[Figure]

Using the resistance model for a spherical particle, this yields

$$\frac{\Delta\gamma}{\gamma} = \frac{\gamma\bar{v}r}{4D_g}\frac{\Delta D_g}{D_g}.$$

George et. al. (1994) calculated the diffusion coefficient of $N_2O_5$ in $N_2$ using the method of Fuller et al. (1966). Back calculated to 273.15 K, they obtained $D_0 = 0.112$ cm$^2$s$^{-1}$. This value is about 24% higher than the diffusion coefficient 0.09 cm$^2$s$^{-1}$ estimated by the LJ-model, which is consistent with our data within the error limits. At 273 K and 25.2 Torr, they obtained $\gamma = 0.020 \pm 0.002$. Assuming a particle radius of 60 $\mu$m, a diffusion coefficient of $D = 3.38$ cm$^2$s$^{-1}$ at 25.2 Torr and $\bar{v} = 231.4$ m s$^{-1}$, the relative error is $\Delta\gamma/\gamma = 0.21\Delta D_g/Dg = 5$%.

To estimate $\gamma$ for the uptake of $ClONO_2$ on ice and subsequent reaction, the diffusion coefficient might be of greater importance. Hanson and Ravishankara (1992) reported for their flow tube reactor study, gas phase diffusion limits transport to the ice surface. They were only able to report an upper limit of $\gamma > 0.3$ since the diffusion coefficient of $ClONO_2$ in He was not accurately known.

We conclude that the diffusion coefficients of $ClONO_2$ and $N_2O_5$ calculated by the LJ-model using $\sigma$ and $\epsilon$ from Patrick and Golden (1983) are a good choice. Therefore, it does not make sense to backward calculate $\sigma$ and $\epsilon$ from our experimental data.

*The concerns about the measurements center around these two molecules and the temperature dependencies of the D's. Both $N_2O_5$'s and $ClONO_2$'s measured T-dependencies differ significantly from that expected for L-J interactions.*

We use the `anova()` test of R (Phillips, 2018) to check if the temperature coefficient $b$ in our Table 3 which we determined for $ClONO_2$ and $N_2O_5$ significantly deviates from the temperature coefficient of the LJ-model. To do this, first a fit is performed using Eq. (5) and Eq. (24) with $D_0$ and $b$ fit parameters. This yields the values given in Table 3. Then the fit is repeated setting $b$ to the value of the Lennard-Jones model. Now the

two fits (statistical models) are compared using the `anova()` test. As you can see, the T-dependencies does not differ significantly from $b$ predicted by Eq. (7):

| Species | Carrier | $b$ (LJ) | $D_0$ [cm$^2$s$^{-1}$] | $a$ [s] | $P$ | Deviation |
|---------|---------|----------|------------------------|---------|-----|-----------|
| ClONO$_2$ | He | 1.72 | $0.307 \pm 0.003$ | – | 0.37 | not significant |
| ClONO$_2$ | N$_2$ | 1.88 | $0.086 \pm 0.001$ | – | 0.08 | weakly significant |
| N$_2$O$_5$ | He | 1.73 | $0.31 \pm 0.01$ | $-12 \pm 3$ | 0.16 | not significant |
| N$_2$O$_5$ | N$_2$ | 1.91 | $0.085 \pm 0.002$ | $-29 \pm 4$ | 0.35 | not significant |

*While losses were addressed, it seems these anomalous T-dependencies suggest there is more to the story. The indirect detection method for these two species is worth some consideration. Hard to come up with reasons why these two molecules interacting with He and N$_2$ should not be describable by L-J potentials.*

We don't think that the titration reaction with NO can affect the temperature dependency because the titration reaction is always performed outside the thermostatted cold box in a heating zone.

Regarding the LJ-model, two things have to be considered:

1. ClONO$_2$ and N$_2$O$_5$ are both polar components with a permanent dipole moment. Therefore, the interactions with He and N$_2$ are not only van der Waals interactions but also dipole – induced dipole interactions. The latter are not considered in the simple LJ-model.

2. For unstable compounds like ClONO$_2$ and N$_2$O$_5$ $\epsilon$ cannot be obtained from viscosity data. $\epsilon$ can only be estimated from the boiling point using Eq. (4). But for polar component this only can be a rough estimation since the boiling point not only depends on the van der Waals interactions but also on dipole – dipole interactions.

**References**

George, C., Ponche, J. L., Mirabel, P., Behnke, W., Scheer, V., and Zetzsch, C.: Study of the Uptake of $N_2O_5$ by Water and NaCl Solutions, J. Phys. Chem., 98, 8780–8784, https://doi.org/10.1021/j100086a031, 1994.

Hanson, D., Ravishankara, A.: The Reaction Probabilities of $ClONO_2$ and $N_2O_5$ on Polar Stratospheric Cloud Materials, J. Geophys. Res., 96, 5081–509, https://doi.org/10.1029/90JD02613, 1991

Hanson, D., Ravishankara, A.: Investigation of the reactive and nonreactive processes involving nitryl hypochlorite and hydrogen chloride on water and nitric acid doped ice, J. Phys. Chem., 96, 2682–2691, https://doi.org/10.1021/j100185a052, 1992

Hu, J.H., Shi, Q., Davidovits, P, Worsnop, D.R., Zahniser, M.S. and Kolb C.E.: Reactive Uptake of $Cl_2$(g) and $Br_2$(g) by Aqueous Surfaces as a Function of $Br^-$ and $Cl^-$ Concentration: The Effect of Chemical Reaction at the Interface, J. Phys. Chem., 99, 8768, https://doi.org/10.1021/j100021a050, 1995.

Phillips, N.D.: YaRrr! The Pirate's Guide to R, https://bookdown.org/ndphillips/YaRrr/comparing-regression-models-with-anova.html , 2018, (last visited February 18, 2020)